# The Impact of *Ulmus macrocarpa* Extracts on a Model of Sarcopenia-Induced C57BL/6 Mice

**DOI:** 10.3390/ijms25116197

**Published:** 2024-06-04

**Authors:** Chan Ho Lee, Yeeun Kwon, Sunmin Park, TaeHee Kim, Min Seok Kim, Eun Ji Kim, Jae In Jung, Sangil Min, Kwang-Hyun Park, Jae Hun Jeong, Sun Eun Choi

**Affiliations:** 1Department of Forest Biomaterials Engineering, Kangwon National University, Chuncheon 24341, Republic of Korea; lgh4107@gmail.com; 2Dr.Oregonin Inc., #802 Bodeum Hall, Kangwondaehakgil 1, Chuncheon 24341, Republic of Korea; kye0519@naver.com (Y.K.); dpkssm0929@naver.com (S.P.); kth02120@naver.com (T.K.); ms23217@naver.com (M.S.K.); 3Industry Coupled Cooperation Center for Bio Healthcare Materials, Hallym University, Chuncheon 24252, Republic of Korea; myej4@hallym.ac.kr (E.J.K.); jungahoo@hallym.ac.kr (J.I.J.); 4Division of Transplantation and Vascular Surgery, Department of Surgery, Seoul National University Hospital, Seoul 03080, Republic of Korea; surgeonmsi@gmail.com; 5Department of Emergency Medical Rescue, Nambu University, Gwangju 62271, Republic of Korea; khpark@nambu.ac.kr; 6Department of Food Science & Biotechnology, Jeonnam State University, Damyang 57337, Republic of Korea; jhjeong@dorip.ac.kr

**Keywords:** *Ulmus macrocarpa* Hance, sarcopenia, muscle apoptosis, muscle atrophy, antioxidant, anti-inflammatory

## Abstract

Aging leads to tissue and cellular changes, often driven by oxidative stress and inflammation, which contribute to age-related diseases. Our research focuses on harnessing the potent anti-inflammatory and antioxidant properties of Korean *Ulmus macrocarpa* Hance, a traditional herbal remedy, to address muscle loss and atrophy. We evaluated the effects of *Ulmus* extract on various parameters in a muscle atrophy model, including weight, exercise performance, grip strength, body composition, muscle mass, and fiber characteristics. Additionally, we conducted Western blot and RT-PCR analyses to examine muscle protein regulation, apoptosis factors, inflammation, and antioxidants. In a dexamethasone-induced muscle atrophy model, *Ulmus* extract administration promoted genes related to muscle formation while reducing those associated with muscle atrophy. It also mitigated inflammation and boosted muscle antioxidants, indicating a potential improvement in muscle atrophy. These findings highlight the promise of *Ulmus* extract for developing pharmaceuticals and supplements to combat muscle loss and atrophy, paving the way for clinical applications.

## 1. Introduction

Skeletal muscle is a significant component of the human body, comprising 40–50% of its mass. However, as we age, the amount of skeletal muscle in our bodies decreases by approximately 1% per year after the age of 30. This decline becomes even more rapid after the age of 65. This gradual loss of muscle tissue, known as sarcopenia, results in reduced muscle size and strength [1]. Also, aging is a biological process that leads to functional and structural changes in tissues and cells over time [2]. These changes can be attributed to oxidative stress, a condition characterized by the excessive generation of chemical substances, such as free radicals, within cells. This can cause damage to the cell’s DNA, proteins, and other important biomolecules, which, in turn, can lead to inflammation. Inflammation is a major mechanism underlying many diseases associated with aging [3]. Inflammatory responses can induce cellular apoptosis and tissue damage, especially in the context of muscles. Cellular apoptosis leads to a reduction in the number of muscle cells, which subsequently diminishes the size and function of the muscles, ultimately causing sarcopenia [4].

Sarcopenia can be caused by various factors, including lack of exercise, increased inflammatory cytokines, elevated production of free radicals or impaired detoxification, low secretion of anabolic hormones, malnutrition, and decreased nerve impulses [5]. The maintenance of muscle mass relies on a balance between protein degradation and synthesis, which is influenced by factors such as hormonal balance, nutritional status, physical activity, exercise, injury, and disease [6].

Muscle decomposition occurs through multiple pathways, including the caspase system pathway, ubiquitin-proteasome-dependent pathway, and autophagy pathway [7]. Sarcopenia involves various genes, such as IGF-1, Atrogin-1, muscle RING-finger protein- 1 (MuRF1), phosphatidylinositol 3-kinase (PI3K)-p85α, Akt, A1R, myostatin, sirtuin 1, and FOXO, which are all involved in muscle protein degradation and synthesis [8,9,10,11,12]. In particular, IGF-1 increases skeletal muscle protein synthesis through the PI3K/Akt/mTOR and PI3K/Akt pathways. The PI3K/Akt pathway can also inhibit the transcription of E3 ubiquitin ligases, which, in turn, suppress FoxO and regulate protein degradation through the ubiquitin proteasome system [12]. Animal models, including those with diabetes and glucocorticoid (GC) administration, have been used to study muscle atrophy [13]. GC administration, specifically dexamethasone, has been widely used to study muscle atrophy and is known to increase reactive oxygen species and suppress the PI3K/AKT signaling pathway, leading to muscle protein degradation [14,15]. Dexamethasone is also used as a representative model of aging in animal experiments [16,17]. Increased levels of GC in blood plasma, resulting from changes in neuroendocrine regulation, contribute to various catabolic systems in muscle, connective tissue, bone, and lymphoid tissue [18]. Prolonged or high-dose use of GC can upregulate the expression of myostatin, E3 ubiquitin ligase, Atrogin-1, and muscle RING-finger protein-1 (MuRF1), ultimately leading to muscle atrophy [19,20,21]. Despite extensive research efforts, there are currently no effective treatments for sarcopenia other than animal protein intake, exercise therapy, and the use of oxymetholone. However, excessive consumption of animal protein can cause kidney and liver damage [22], and long-term use of oxymetholone is associated with various side effects, including liver toxicity, hair loss, and acne [23]. During high-intensity physical activity, muscle damage occurs, leading to an increase in muscle enzymes such as creatine kinase and lactate dehydrogenase, which serve as biomarkers of muscle damage in the blood [24]. Given the side effects of current treatments, there is an urgent need for research and development of new materials derived from natural products that are safe for the human body and demonstrate excellent activity.

The *Ulmus* genus has been used in oriental and traditional remedies in Korea for various symptoms throughout history, including as an anti-inflammatory, antiulcer, anticancer, anthelmintic, antifungal, pain relief, and diuretic, and used to treat insomnia, edema, fever, and scabies [25]. Specifically, *Ulmus macrocarpa* Hance, a member of the *Ulmus* genus, is commonly used in traditional Korean medicine. The inner bark is known as “yu-baek-pi (楡白皮)”, and the dried root bark is called “yu-geun-pi (楡根皮)”. Known for its non-toxic nature, it has been used to treat conditions such as swelling and arthritis, and as an anti-inflammatory agent [25]. Furthermore, *Ulmus macrocarpa* Hance has a long history of being used for treating infections in China, Japan, Korea, and India, with documented methods of usage [26]. Recent studies have further demonstrated the remarkable anti-inflammatory and antioxidant effects of *Ulmus macrocarpa* Hance extracts [27,28]. Ongoing research continues to explore its traditional medicinal use, encompassing investigations into apoptosis regulation [29] and osteoporosis prevention [26]. Consequently, there has been increasing attention to the potent antioxidant and anti-inflammatory properties of *Ulmus macrocarpa* Hance extract, along with its confirmed role in regulating cell apoptosis. As a result, efficacy evaluation experiments were conducted on models of age-related muscle loss and muscle atrophy.

Previous research has identified a catechin 7-O-*β*-D-apiofuranoside as a standard compound of the *Ulmus* genus [28,30]. It has also been established that *Ulmus* extract has antioxidant properties [31] and anti-inflammatory properties [32]. More recent studies have reported that extracts from the *Ulmus* genus exhibit excellent biological activities, including anticancer effects [33,34], activation and regulation of immune function [35], and apoptosis-regulating effects on human papilla cells [36].

Our research team’s primary objective is to develop natural medicinal products for muscle loss and muscle atrophy. Additionally, we are conducting research on and development of health functional foods. To obtain final approval from the Ministry of Food and Drug Safety, both in vitro and in vivo pre-clinical stages are necessary. Therefore, we conducted pilot studies on C2C12 cells, which are muscle myoblast cells commonly used in in vitro mechanism studies related to muscle tissue.

In these pilot scale studies, we have successfully determined the optimal conditions for extracting high-purity catechin 7-O-*β*-D-apiofuranoside, a standard compound found in the *Ulmus* genus. We have also confirmed the excellent activity of both the *Ulmus* genus extract and catechin 7-O*-β*-D-apiofuranoside in terms of cell viability, antioxidant activity, expression of the apoptosis inhibitor Bcl-2, reduction of apoptosis-inducing factors caspase-3 and PARP, decrease in muscle decomposition factors Atrogin1 and MuRF1, and expression of muscle synthesis factors Myogenin and MyoD [37]. However, while these results are promising in the controlled in vitro stage, it is important to note that the more complex in vivo stage may yield different outcomes due to various biological interactions [38]. That is why in vivo experiments are deemed necessary for pre-clinical studies on drug development, research, and the creation of health functional foods. In our in vivo experimental stage, our research team used the well-known corticosteroid medication dexamethasone to establish an animal model of muscle loss and atrophy induced by aging in 8-week-old C57BL/6 mice [39].

The dexamethasone-induced animal model was used to investigate changes in protein expression related to muscle mass, relative muscle mass, skeletal muscle cross-sectional area, muscle cell death, and alterations in protein expression associated with muscle synthesis and degradation. Additionally, we performed specific verification of protein expression and mRNA gene expression of biomarkers associated with muscle formation and atrophy, such as Myogenin, MyoD, IGF-1, Myostatin, Atrogin1, MuRF1, as well as inflammation-related biomarkers IL-1β, IL-6, TNF-α, and antioxidant-related biomarkers SOD, catalase, and GPX, during the in vivo stage. Our research team aimed to conduct pre-clinical research in four distinct categories: inhibiting oxidative stress, anti-apoptosis, promoting muscle protein synthesis, and inhibiting muscle protein degradation, to explore the effectiveness and mechanisms of muscle loss and atrophy.

## 2. Results

### 2.1. Phytochemical Analysis

#### 2.1.1. Qualitative Analysis of UME (TLC)

Through thin layer chromatography (TLC), a qualitative analysis of UME was conducted. By comparing the Rf values and color reactions with the standard sample of catechin 7-O-*β*-D apiofuranoside, it was determined that UME is the same substance as the standard sample (Figure 1).

#### 2.1.2. Measurement of the Molecular Weight of UME (LC-MS/MS)

The samples were analyzed using LC-MS/MS. The spectrum was measured in negative mode, and it was confirmed that it had a molecular weight of 422.38 g/mol (Figure 2), which is the same as that of the catechin 7-O-*β*-D-apiofuranoside, which is known as an indicator material for *Ulmus* species.

#### 2.1.3. Quantitative Chromatographic Analysis of UME

The relationship between the concentration and the peak-area was measured by the minimum square method (R^2^ value). The standard calibration curve obtained with concentrations in four increments was Y = 11344X + 154313 (R^2^ = 0.999), as shown in Figure 3A. A good linearity (correlation coefficient 0.999) was shown for the calibration curve. The retention time of 77-O-*β*-D-apiofuranoside was 12.58 ± 0.01 min (Figure 3B). The average content of 7-O-*β*-D-apiofuranoside in UME was calculated as 135.98 ± 0.03 (*n* = 3) by the above formula (Figure 3C).

### 2.2. Effect on Body Weight of Experimental Animals

The weights of the experimental animals were measured once a week throughout the test period and are presented in Table 1. All the animals in the test groups continued to gain weight during the experiment, showing normal weight changes. However, starting from the third week, the MAC groups (G2, G3, G4, and G5) experienced a significant decrease in body weight compared to the NC group (G1). This decrease is believed to be a result of muscle loss induced by the DEX injection. Within the MAC groups (G2, G3, G4, and G5), there were no significant differences in the body weight of the animals among the test groups (Table 1).

### 2.3. Effect on Exercise Time and Exercise Capacity

To assess the influence of UME on the exercise performance of the experimental animals, the exercise regimen commenced with a 5 min session at a 10-degree incline and a speed of 10 m/min. Subsequently, the exercise intensity was incremented by increasing the speed by 1 m/min every minute. The duration of exercise until exhaustion, with a speed of up to 25 m/min, was recorded and is presented in Table 2. The exercise duration for the NC group (G1) was 29.2 ± 2.3 min, the longest among all groups tested. In contrast, the exercise duration until exhaustion in the MAC group (G2) was 15.1 ± 0.9 min, significantly shorter compared to the NC group (G1). Among the muscle loss test groups (G2, G3, G4, and G5), the UME-treated group receiving 100 mg/kg BW (G4) exhibited an exercise duration of 19.6 ± 1.1 min, while the group receiving 200 mg/kg BW (G5), serving as the muscle loss control group, showed a duration of 22.4 ± 0.9 min. This represents a significant increase in exercise duration compared to G2 (Table 2). Furthermore, the analysis of exercise output revealed a significant reduction to 562.4 ± 45.6 J in the muscle loss control group (G2) compared to 1595.9 ± 173.8 J in the NC group (G1). However, the exercise output significantly increased to 806.4 ± 72.2 J and 994.1 ± 66.3 J in the 100 mg/kg BW UME-treated group (G4) and 200 mg/kg BW UME-treated group (G5), respectively (Table 2 and Figure 4).

### 2.4. Effects on Grip Strength

To assess the impact of UME administration on the grip strength of experimental animals, we utilized a grip strength tester specifically designed for small animals (BIO-GS3 Grip strength test, Chaville, France). As depicted in Figure 4, the grip strength of the MAC group (G2) was significantly decreased to 76.3 ± 3.6 g compared to 121.0 ± 8.4 g in the NC group (G1). Notably, this reduction was mitigated by the administration of 200 mg/kg BW UME, with grip strength significantly increasing to 87.0 ± 2.5 g.

### 2.5. Changes in Body Fat Percentage and Lean Body Mass Percentage

The impact of UME administration on lean body mass and body fat percentage was examined by measuring body composition. The results are presented in Table 2. The MAC group (G2) showed a significant increase in body fat percentage to 19.85 ± 0.54%, compared to 15.29 ± 0.43% in the NC group (G1). Although there was a tendency for the increase in body fat percentage to decrease with the administration of UME, no significant difference was observed. The MAC group (G2) also exhibited a significant decrease in the percentage of lean body mass to 80.15 ± 0.54%, in comparison to 84.71 ± 0.43% in the NC group (G1). While there was a tendency for the levels to increase with UME administration, no significant difference was observed.

### 2.6. Muscle Weight Changes

After concluding the test, we measured the muscle weight of each part of the experimental animals, and the results are presented in Table 3. Compared to the NC group (G1), the weights of the QF, GA, SOL, EDL, and TA significantly decreased in the MAC group (G2). In the group treated with 100 mg/kg BW UME (G4), the weight of the SOL and EDL significantly increased compared to the MAC group (G2), while in the group treated with 200 mg/kg BW UME (G5), the weight of the QF, SOL, EDL, and TA significantly increased. To account for the decrease in body weight caused by muscle atrophy, we calculated the weight of each muscle per 100 g of body weight, which is presented in Table 3. Compared to the NC group (G1), the relative weights of the QF, GA, EDL, and TA significantly decreased in the MAC group (G2). In the group treated with 100 mg/kg BW UME (G4), the relative weight of the SOL and EDL significantly increased compared to the MAC group (G2), while in the group treated with 200 mg/kg BW UME (G5), the relative weight of the QF, SOL, EDL, and TA significantly increased.

### 2.7. Effects on Muscle Fiber Cross-Sectional Area of the Tibialis Anterior (TA)

Muscle size is regulated by intracellular signaling processes that can either promote muscle growth (anabolism) or muscle breakdown (catabolism). When the signaling responses that lead to muscle protein degradation increase, it can result in muscle loss and a decrease in muscle size or the number of muscle fibers. In this study, the TA muscles were used for H and E staining to observe and measure the cross-sectional area (CSA) of the muscle fibers, which provides an indication of the extent of muscle loss. As shown in Figure 5, a significant decrease in the CSA of muscle fibers was observed in the MAC group (G2) compared to the NC group (G1). However, there was a slight increase in the CSA size of muscle fibers in the test groups for muscle atrophy that received UME compared to the MAC group (G2). The CSA measurements revealed that the MAC group (G2) had a significantly lower CSA (3547 ± 344) compared to the NC group (G1) with a CSA of 8931 ± 918. Furthermore, the groups treated with 100 mg/kg BW UME (G4) and 200 mg/kg BW UME (G5) showed a significant increase in the decrease of CSA. These findings indicate that UME has significant potential for promoting recovery from muscle atrophy (Figure 5).

### 2.8. Effects on Protein Expression in Gastrocnemius

#### 2.8.1. Effects on Signaling Pathways Regulating Muscle Protein Synthesis and Degradation

Muscle protein synthesis is a crucial anabolic process involved in muscle formation and regeneration. It is facilitated through the insulin-like growth factor-1/PI3K/Akt signaling pathway. This pathway activates the PI3K and Akt signals, which then regulate mTOR to promote protein synthesis [12]. On the other hand, muscle protein degradation is regulated by signaling pathways involving the forkhead box O (FoxO) and the ubiquitin-proteasome system. In skeletal muscle, there are three isoforms of FoxO (FoxO1, FoxO3a, and FoxO4), and they are predominantly located in the nucleus [40]. When Akt phosphorylates FoxO, it becomes inactivated and moves from the nucleus to the cytoplasm. Conversely, decreased Akt activation leads to FoxO activation in the nucleus, which promotes the expression of muscle atrophy genes (atrogin-1, MuRF1) and subsequent protein degradation [41].

In this study, we conducted Western blot analysis using muscle tissue lysate (specifically gastrocnemius, GA) to investigate the effect of UME on the activation of Akt, mTOR, and FoxO3α. We observed a significant decrease in the expression of p-Akt in the MAC group (G2) compared to the NC group (G1). However, in the muscle atrophy test groups (G2, G3, G4, and G5), the expression of p-Akt significantly increased in the 100 mg/kg BW UME-treated group (G4) and the 200 mg/kg BW UME-treated group (G5) compared to the MAC group (G2). The expression of total Akt showed no significant difference between the NC group (G1) and the MAC group (G2). But, in the muscle atrophy test groups (G2, G3, G4, and G5), the expression of Akt significantly decreased in the 200 mg/kg BW UME-treated group (G5) compared to the MAC group (G2). To assess Akt activity, we calculated the p-Akt/Akt ratio, and it showed a significant decrease in Akt activity in the MAC group (G2) compared to the NC group (G1). This decrease was also significant in the 50 mg/kg BW UME-treated group (G3). On the other hand, Akt activity significantly increased in the 100 mg/kg BW UME-treated group (G4) and the 200 mg/kg BW UME-treated group (G5) compared to the NC group (G1).

According to Figure 6 the expression of p-mTOR showed a significant decrease in the MAC group (G2) compared to the NC group (G1). In the groups subjected to the muscle atrophy test (G2, G3, G4, and G5), the expression of p-mTOR significantly increased in all groups that received the test substance (G3, G4, and G5) compared to the MAC group (G2). The expression of mTOR also showed a significant decrease in the MAC group (G2) compared to the NC group (G1). This decrease was further increased significantly in all groups that received the test substance (G3, G4, and G5). The evaluation of mTOR activity, indicated by the ratio of p-mTOR/mTOR, revealed a significant decrease in mTOR activity in the MAC group (G2) compared to the NC group (G1). This decrease was further mitigated significantly in the groups that received UME (G3, G4, and G5).

The expression of p-FoxO3α, as shown in Figure 6, did not differ significantly among the experimental groups (G1, G2, G3, G4, and G5). However, the expression of FoxO3α was significantly higher in the MAC group (G2) compared to the NC group (G1). In the muscle atrophy test groups (G2, G3, G4, and G5), the expression of FoxO3α was significantly lower in the 100 mg/kg BW UME-treated group (G4) and 200 mg/kg BW UME-treated group (G5) compared to the MAC group (G2). The ratio of p-FoxO3α/FoxO3α was significantly lower in the MAC group (G2) compared to the NC group (G1). This decrease was significantly greater in the group treated with 200 mg/kg BW UME (G5).

Combining the above findings, Akt and mTOR activities were significantly diminished by muscle loss induction. However, UME administration effectively restored Akt and mTOR activities. Furthermore, the FoxO3α ratio was markedly elevated in the sarcopenic control group (G2) compared to the NC group (G1), yet notably reduced in the group receiving 200 mg/kg BW UME administration (G5). This underscores UME’s role in enhancing the expression of AKT and mTOR, pivotal factors in protein synthesis, while concurrently diminishing the expression of FoxO3α, a protein degradation factor.

#### 2.8.2. Effect on Expression of Apoptosis Regulatory Proteins

Apoptosis is a crucial biological process that plays a vital role in tissue regeneration, maintaining homeostasis, and promoting growth. It becomes a significant factor in muscle atrophy caused by chronic inflammation and oxidative stress. During apoptosis, the pro-apoptotic factor Bax is activated and increases from the cell membrane, while the expression of the anti-apoptotic factor Bcl-2 is inhibited. This imbalance disrupts mitochondrial function, leading to the release of cytochrome c into the cytoplasm, which triggers apoptosis [42]. In this study, we examined the effect of UME on the expression of Bcl-2 and Bax using Western blot analysis on muscle tissue lysate (gastrocnemius, GA). Figure 7 shows that the expression of Bcl-2 significantly decreased in the MAC group (G2) compared to the NC group (G1). However, there were no significant differences among the UME-treated groups (G3, G4, and G5) when compared to the MAC group (G2). On the other hand, the expression of Bax significantly increased in the MAC group (G2) compared to the G1 group. In the muscle atrophy test groups (G2, G3, G4, and G5), the expression of Bax significantly decreased in the 100 mg/kg BW UME-treated group (G4) and the 200 mg/kg BW UME-treated group (G5) compared to the MAC group (G2) (Figure 7).

### 2.9. Effects on mRNA Expression in Muscle 

#### 2.9.1. Effects on Muscle Degradation and Formation-Related Gene Expression

The metabolism associated with muscle atrophy prominently highlights the key mechanism known as the ubiquitin-proteasome signaling pathway. When signals related to muscle atrophy occur, the proteins that make up the skeletal muscles undergo ubiquitination. This process involves ubiquitin, ubiquitin-conjugating enzyme (E2), ubiquitin-protein ligases (E3), and proteasome factors, ultimately leading to the progression of muscle atrophy [43]. In this process, two important E3 ubiquitin ligases, muscle atrophy F-box/atrogene-1 (MAFbx) and muscle RING-finger protein-1 (MuRF1), are specifically expressed in muscles. These ligases play a crucial role in activating protein ligases, thereby accelerating muscle atrophy [44]. Additionally, myostatin, a member of the TGF-β family, serves as a muscle cell-specific inhibitory substance that is primarily expressed and secreted in muscles. Myostatin inhibits the growth of skeletal muscles by reducing Akt/mTOR/p70S6K signaling [45]. However, exercise plays a role in muscle regeneration and formation. During this process, regulatory factors called myogenin are activated, which are involved in muscle formation. These factors, including MyoD, Myf5, Myogenin, and Mrf4, make up a basic helix–loop–helix transcription factor in muscle and are expressed in satellite cells. Among them, MyoD determines the muscle progenitors, known as myoblasts, and Myogenin promotes muscle differentiation [46]. In this study, we examined the impact of inducing muscle atrophy and administering substances on the expression of genes related to muscle formation (MyoD, Myogenin) and genes related to muscle atrophy (Myostatin, Atrogin1, MuRF1) in the SOL muscle. The results are presented in Figure 8 and Table 4.

The expression of genes related to muscle formation, including MyoD, Myogenin, and IGF-1 mRNA, decreased significantly in the MAC group (G2) compared to the NC group (G1). However, in the muscle atrophy treatment groups, the expression of MyoD, Myogenin, and IGF-1 mRNA increased significantly compared to the MAC group (G2), particularly in the 200 mg/kg BW UME-treated group (G5). The expression of muscle atrophy-related genes, such as Myostatin, Atrogin1, and MuRF1(muscle RING finger protein 1) mRNA, increased significantly in the MAC group (G2) compared to the NC group (G1). However, in the muscle atrophy treatment groups (G2, G3, G4, and G5), the expression of Myostatin, Atrogin1, and MuRF1 mRNA decreased significantly compared to the control group for muscle atrophy (G2), especially in the 200 mg/kg BW UME-treated group (G5).

#### 2.9.2. Effects on Inflammation-Related Rene Expression

Pro-inflammatory cytokines, including IL-1β, IL-6, and TNF-α, play a crucial role in regulating the response of activated macrophages [47]. In this study, we examined the mRNA levels of inflammation-related genes IL-1β, IL-6, and TNF-α in the soleus muscle. The results are presented in Figure 9 and Table 4. As depicted in Figure 9 and Table 4, the expression levels of IL-1β, IL-6, and TNF-α mRNA in the soleus muscle were significantly higher in the MAC group (G2) than in the NC group (G1). Among the experimental groups with muscle atrophy (G2, G3, G4, and G5), the expression levels of IL-1β and IL-6 mRNA significantly decreased in the group treated with 200 mg/kg BW UME (G5) compared to the MAC group (G2). However, the expression of TNF-α mRNA showed a tendency to decrease with UME administration, but the difference was not statistically significant (Figure 9 and Table 4).

#### 2.9.3. Effects on Antioxidant-Related Gene Expression

Oxidative stress is one of the key factors causing skeletal muscle atrophy; it is activated in the early stages of muscle atrophy and can be regulated by various factors [48]. This study analyzed the levels of mRNA for antioxidant genes SOD2, catalase, and GPx1 in the soleus muscle. The results are presented in Figure 10 and Table 4. The mRNA expressions of SOD2, catalase, and GPx1 significantly decreased in the MAC group (G2) compared to the NC group (G1). In the groups where muscle atrophy was induced (G3, G4, and G5), the expression of GPx1 mRNA significantly increased in the group treated with 100 mg/kg BW UME (G4) and the group treated with 200 mg/kg BW UME (G5) compared to the MAC group (G2).

In comparison to the MAC group (G2), the expressions of SOD2 and catalase mRNA showed a tendency to increase with the administration of UME, but this increase was not statistically significant.

## 3. Discussion

Previous studies have found that plants belonging to the *Ulmus* genus have various significant biological activities. These activities include antioxidant, anti-inflammatory, and immunomodulatory effects, as well as the inhibition of angiogenesis and regulation of apoptosis in human papilla cells [24,31,32,33,35]. In a prior study, our research team confirmed the remarkable activity of *Ulmus* extracts and catechin 7-O-*β*-D-apiofuranoside against muscle loss. We examined the protein expression and mRNA gene expression of muscle formation and atrophy-related biomarkers, such as Myogenin, MyoD, IGF-1, Myostatin, Atrogin1, and MuRF1, at the in vitro stage. Additionally, we successfully optimized a pilot-scale *Ulmus macrocarpa* Hance extraction method for catechin 7-O*-β*-D-apiofuranoside, which is a standard component of the *Ulmus* genus [36]. In this study, at the in vivo stage, we specifically confirmed various parameters, including body weight, grip strength, muscle mass, CSA, exercise duration, and the protein expression and mRNA gene expression of Myogenin, MyoD, IGF-1, Myostatin, Atrogin1, and MuRF1. Furthermore, we examined inflammatory biomarkers such as IL-1β, IL-6, TNF-α, and antioxidant biomarkers like SOD, catalase, and GPX.

Compared to the NC group, the experimental animals in the muscle loss control group showed a significant decrease in body weight. However, within the muscle loss test group, the administration of UME did not affect the body weight of the experimental animals, confirming that it was due to fat content. Muscle atrophy resulted in a significant decrease in the relative weight of the QF, GA, EDL, and TA muscles. The administration of UME significantly increased the relative weight of the QF, SOL, EDL, and TA muscles. The group with the highest concentration, G5, showed the most significant recovery. When the cross-sectional area of the TA muscles was stained and compared with the NC group, muscle atrophy induction significantly reduced the size of the muscle fiber cross-sectional area. In the muscle loss test group, the administration of UME effectively increased the size of the muscle fiber cross-sectional area in a concentration-dependent manner compared to the muscle loss control group. The duration of exercise (time to exhaustion) and exercise volume significantly increased in the G4 and G5 groups, which received the test substance, compared to the decreased exercise time and volume in the muscle loss control group. Grip strength, which decreased in the muscle loss control group (G2), significantly increased in G5, where the highest concentration of UME was administered.

Muscle protein synthesis is a vital anabolic process that plays a key role in muscle formation and regeneration. It is facilitated by the insulin-like growth factor-1/PI3K/Akt signaling pathway. This pathway activates PI3K and Akt signals, which in turn regulate mTOR to promote protein synthesis [12]. This study used Western blot analysis with muscle tissue lysates to examine the impact of UME administration on the activation of Akt, mTOR, and Foxo3α. The activation of Akt and mTOR in the muscle tissue significantly decreased due to the induction of muscle atrophy in GA. However, the administration of UME significantly increased the decreased activation of Akt and mTOR induced by muscle atrophy. In particular, G5, which received 200 mg/kg BW of UME, showed a nearly identical recovery to the NC group.

Apoptosis is a vital biological process that plays a key role in tissue regeneration, maintaining homeostasis, and promoting growth. It becomes a significant factor in muscle atrophy caused by factors like chronic inflammation and oxidative stress [44]. In the muscle loss control group, the expression of Bcl-2 significantly decreased compared to the NC group. However, when comparing the muscle loss test group with the muscle loss control group, no significant differences were observed in any of the groups administered with UME. In the muscle loss control group, the expression of Bax increased significantly compared to the NC group. On the other hand, in the muscle loss test group, a significant decrease in the expression of Bax was observed in groups G4 and G5 compared to the muscle loss control group.

The regulatory factors responsible for muscle formation, such as myogenin, become activated with the muscle atrophy inducement. These factors, collectively referred to as MyoD, Myf5, Myogenin, and Mrf4, form a basic helix–loop–helix transcription factor in muscle and are expressed in satellite cells.

Among these factors, MyoD plays a crucial role in determining the muscle progenitors, known as myoblasts, while Myogenin triggers muscle differentiation [46]. When UME was administered to the SOL, it led to an increase in the expression of mRNAs related to muscle formation such as MyoD, Myogenin, and IGF-1. These mRNA levels had previously decreased due to muscle loss induction. Increasing the expression of myostatin, E3 ubiquitin ligase, Atrogin1, and muscle RING finger protein 1 (MuRF1) ultimately leads to muscle atrophy [18]. The administration of UME decreased the expression of mRNAs associated with muscle atrophy, specifically Myostatin, Atrogin1, and MuRF1. These mRNAs had previously increased due to muscle loss. This confirmed that UME has a significant impact on different pathways involved in muscle loss and atrophy, including the promotion of protein synthesis and the inhibition of protein breakdown.

Pro-cytokines such as IL-1β, IL-6, and TNF-α regulate the participation of molecules primarily produced by activated macrophages [47]. In the SOL group, mRNA expression of IL-1β, IL-6, and TNF-α increased due to muscle atrophy induction. G5, which received 200 mg/kg BW of UME, showed significantly reduced expression of the inflammatory-related mRNAs IL-1β and IL-6, which were increased by muscle atrophy induction.

Oxidative stress is a key factor in causing skeletal muscle atrophy. It is activated early in the development of muscle atrophy and can be influenced by various factors [48]. The body possesses an antioxidant system consisting of different substances and enzymes, including SOD, CAT, and GPx. SOD converts oxygen free radicals into hydrogen peroxide (H_2_O_2_). Then, CAT and GPx convert hydrogen peroxide into water, effectively detoxifying the substance [49].In the SOL, the expression of SOD2, catalase, and GPx mRNA decreased when muscle atrophy was induced. However, when 200 mg/kg of UME was administered, the decreased expression of GPx mRNA induced by muscle atrophy significantly increased. This study confirms the effectiveness of UME in preventing and reducing oxidative stress. In this study, we have conducted a thorough pre-clinical investigation on the effectiveness and underlying mechanisms of four established categories: reducing oxidative stress, preventing muscle cell death, promoting muscle protein synthesis, and inhibiting muscle protein breakdown. Our findings have allowed us to move forward to the clinical stage, indicating that these results can be used as crucial elements in the development of pharmaceuticals and health supplements to prevent muscle loss and atrophy.

## 4. Materials and Methods

### 4.1. Plant Extract Materials

This study was conducted using *Ulmus macrocarpa* Hance extract (UME: Lot. No. DJTT-06789; DanjoungBio Co., Ltd., Wonju, Republic of Korea). The *Ulmus macrocarpa* Hance bark used in the study was purchased from Seoul Yakryeong Market and we obtained certification from Professor Choi (Department of Forest Biomaterials Engineering, Kangwon National University). After cleaning and washing to remove impurities, the bark was used as an experimental material. The sample of *Ulmus macrocarpa* Hance extract (UME: Lot. No. DJTT-06789) is stored at the Department of Forest Biomaterials Engineering, Kangwon National University. The *Ulmus macrocarpa* Hance extract (UME: Lot. No. DJTT-06789) was prepared with 50% edible ethanol concentration for 8 h at 80 °C. Dextrin, an excipient, was added at a concentration of 10%. The extraction process was conducted by DanjoungBio Co., LTD (Wonju, Republic of Korea) [38].

### 4.2. Phytochemical Analysis

#### 4.2.1. Qualitative Analysis of UME (TLC)

Thin layer chromatography was performed to identify the *Ulmus macrocarpa* extract (UME). A standard sample and *Ulmus macrocrapa* extract (UME) were each taken at 2.5 mg and dissolved in 1 mL of methanol to prepare 2500 ppm samples. Using a silica gel plate, the two types of samples were spotted, and the plate was developed with a mobile phase consisting of chloroform/methanol/water (CMW) in a ratio of 70:30:4. After completely drying the developed silicagel plate, it was observed at 254 nm through a UV lamp, and analysis was performed using three coloring reagents: 10% H_2_SO_4_, p-anisaldehyde H_2_SO_4_, and FeCl_3_.

#### 4.2.2. Measurement of the Molecular Weight of UME (LC-MS/MS)

LC-MS/MS analysis was performed using UME, and the AB SCIEX (QTRAP 4500, Billerica, MA, USA) model was used as the equipment. Column conditions used a combination of a SkyPak C18 column (5 μm) and a Phenomenex KJ0-4282 guard column. The mobile phases used were 0.9% acetic acid in water (A) and acetonitrile (B), the wavelength was analyzed at 280 nm, and the analysis was conducted at a flow rate of 1 mL/min for 35 min. The gradient program was as follows: 0–0 min, 5% B; 0–5 min, 10% B; 5–7 min, 15% B; 7–30 min, 25% B; 30–31 min, 5% B; 31–35 min, 5% B.

#### 4.2.3. Quantitative Chromatographic Analysis of UME (HPLC)

HPLC analysis of *Ulmus macrocarpa* Hance extract was performed using a ChroZen HPLC system and a ChroZen HPLC UV/Vis Detector (Young In Chromass, Anyang, Republic of Korea). Column conditions were used in combination with a SkyPak C18 column (5 μm) and a Phenomenex KJ0-4282 guard column. The inject volume was 20 μL, and the wavelength was 280 nm. The analysis was conducted at a flow rate of 1 mL/min for 35 min using 0.9% acetic acid in water (A) and acetonitrile (B) as the mobile phases. The gradient program was as follows: 0–0 min, 5% B; 0–5 min, 10% B; 5–7 min, 15% B; 7–30 min, 25% B; 30–31 min, 5% B; 31–35 min, 5% B. For the quantitative analysis of the *Ulmus macrocarpa* Hance extract, catechin 7-O*-β*-D-apiofuranoside that was previously isolated from *Ulmus macrocarpa* Hance was used as the standard material. Catechin 7-O*-β*-D-apiofuranoside was weighed to 1 mg and dissolved in 1 mL of methanol. The dissolved standard solution was diluted to 1000, 500, 250, 125, 62.5, and 31.25 ppm, respectively.

### 4.3. Ethical Statement and Animals

The experimental animal guidelines followed the regulations for the use and care of experimental animals set by the Korea Food and Drug Administration, and approval for the use of experimental animals was obtained from the Institutional Animal Care and Use Committee of Hallym University, approved by the National Agricultural Products Quality Management Service under the Ministry of Agriculture, Food and Rural Affairs of Korea (approval number: Hallym 2022-59).

We obtained specific pathogen-free (SPF) male C57BL/6 mice, aged 8 weeks, from Dooyeol Biotech Co. Ltd. in Seoul, Korea. The mice were housed in the environmentally controlled SPF animal facility at Hallym University, where the temperature was maintained at 23 ± 3 °C and the relative humidity at 50 ± 10%. They were kept on a 12 h light/dark cycle, and had access to a non-purified commercial rodent diet and tap water ad libitum.

### 4.4. Experimental Design and Treatment 

After one week of acclimation, C57BL/6 mice (9-week-old, male, 23.7 g) were divided randomly into five groups with ten mice in each group: (G1) the normal control group (NC), (G2) the muscle atrophy control group (MAC), (G3) the muscle atrophy group treated with 50 mg/kg body weight (BW) per day of UME (MA + UME50), (G4) the muscle atrophy group treated with 100 mg/kg BW per day of UME (MA + UME100), and (G5) the muscle atrophy group treated with 200 mg/kg BW per day of UME (MA + UME200). The mice in each group received UME (100 μL/mouse) dissolved in sterile water oraly once a day for four weeks. The mice in the NC and MAC groups were given an equal volume of sterile water as a vehicle by oral gavage. Two weeks after the UME treatment, all mice except those in the NC group were injected intraperitoneally with dexamethasone (Merck & Co., Inc., Kenilworth, NJ, USA) at a dosage of 5 mg/kg BW per day for 14 days to induce muscle atrophy. The mice in the NC group were injected with an equal volume of saline instead of dexamethasone (Figure 11). One day before the end of the experiment, animals were anesthetized, and body composition was measured using dual-energy X-ray absorptiometry (DEXA, PIXImusTM, GE Lunar) to assess lean body mass and body fat percentage. At the end of the experiment, the mice were anesthetized with tribromoethanol diluted with tertiary amyl alcohol. The mice were then euthanized by cervical dislocation, and the skeletal muscles (quadriceps femoris muscle [QF], gastrocnemius muscle [GA], soleus muscle [SOL], extensor digitorum longus muscle [EDL], and tibialis anterior muscle [TA]) were excised and weighed. Each muscle was stored at −70 °C until further analysis.

### 4.5. Measurement of Exercise Capacity 

In order to assess exercise capacity, we conducted the exhaustive treadmill test using a rodent treadmill (Exer-3R Treadmill; Columbus Instruments, Columbus, OH, USA) as previously described [50]. Three days before the end of the experiment, the mice were individually placed on the treadmills at a 10-degree incline and warmed up by running at a speed of 10 m/min for 5 min. After the warm-up, the speed was increased by 1 m/min every minute until reaching 25 m/min. The point of exhaustion was determined when the mice remained in constant contact with the shock grid for 10 s. The exhaustion time was recorded, and the exercise capacity was calculated using the following formula: exercise capacity (J, kg × m^2^ × s^−2^) = body weight (kg) × speed (m/s) × time (s) × grade × 9.8 (m/s^2^).

### 4.6. Measurement of Grip Strength

To assess skeletal muscle strength, we conducted a grip strength test using a grip strength meter (BIO-G53; Bioseb, Chaville, France). On the day prior to the end of the experiment, the mice gripped the T-bar of the meter with their forelimb and their tail was pulled at a constant speed of 2 cm/s. The maximum force exerted just before the mice released their grip was measured. We took five measurements for each mouse and calculated the average to determine the skeletal muscle strength, which was expressed in grams.

### 4.7. Histological Analysis 

The harvested TA was embedded in Tissue-Tek optical cutting temperature (OCT) compound (Sakura Finetech, Tokyo, Japan) on a cryo mold and immediately frozen. The frozen tissues were then sectioned into 6 mm thick slices using a cryo-microtome (MICROM HM 520; Thermo Scientific, Waltham, MA, USA) at −25 °C and stained with hematoxylin and eosin (H and E). The stained tissues were examined and photographed under a light microscope (AxioImager; Carl Zeiss, Jena, Germany). The cross-sectional areas (CSAs) of the muscle fibers were quantified using Image J software (Version 1.54h).

### 4.8. Western Blot Analysis

The harvested GA tissue was lysed, and Western blot analyses were performed following the methods described in a previous study [51]. Antibodies against Akt, phopho-Akt, FoxO3a, phospho-FoxO3a, mTOR, phospho-mTOR, Bcl-2, Bax, and β-actin were obtained from Cell Signaling Technology (Beverly, MA, USA). Each protein band was detected using Luminata Forte Western HRP Substrate (Millipore, Billerica, MA, USA). The relative expression of the target protein was analyzed using an ImageQuantTM LAS 500 imaging system (GE Healthcare Bio-Science AB, Uppsala, Sweden) and normalized to β-actin.

### 4.9. Real-Time Reverse Transcription Polymerase Chain Reaction (RT-PCR) Analysis

The harvested GA tissue (50 mg) were homogenized with a polytron tissue homogenizer in 750 μL ice-cold lysis buffer (20 mM HEPES, pH 7.5, 150 mM NaCl, 1 mM EDTA, 1 mM EGTA, 100 mM NaF, 10 mM sodium pyrophosphate, and 1% Triton X−100) containing 5 mM iodoacetic acid and protease inhibitor (iNtRON Biotechnology, Inc., Seongnam, Republic of Korea), then solubilized at 4 °C for 40 min. Insoluble material was removed by centrifugation at 14,800× *g* for 10 min and the supernatant was collected and used for Western blot analyses. The protein contents of the lysates were measured using a BCA protein assay kit (Thermo Scientific, Rockford, IL, USA). Western blot analyses were conducted as described previously [50]. The antibodies against Akt (1:1000, Cat No. 9272), phospho-Akt (1:1000, Cat No. 4051), FoxO3 (1:1000, Cat No. 2497), phospho-FoxO3 (1:1000, Cat No. 9466), mTOR (1:1000, Cat No. 2972), phospho-mTOR (1:1000, Cat No. 5536), Bcl-2 (1:1000, Cat No. 3498), Bax (1:1000, Cat No. 2772), and β-actin (Cat No. 3700) were obtained from Cell Signal Technology (Beverly, MA, USA). Anti-rabbit IgG HRP-linked antibody (1:5000, Cat No. 7074) and anti-mouse IgG HRP-linked antibody (1:5000, Cat No. 7076) were obtained from Cell Signal Technology. Each protein band was detected using Luminata Forte Western HRP Substrate (Millipore, Billerica, MA, USA). The relative expression of the target protein was analyzed using an ImageQuant^TM^ LAS 500 imaging system (GE Healthcare Bio-Science AB, Uppsala, Sweden) and normalized to β-actin.

The SOL tissue was harvested and total RNA was extracted. Real-time PCR was performed using a QuantiNova SYBR Green kit (Qiagen, Valencia, CA, USA) and Rotor-Gene 3000 PCR (Corbett Research, Mortlake, Australia), following the method described by Lee et al. in 2019. The primer sequences used for the real-time PCR analysis in this study can be found in Table 5. The relative expression of the target mRNA was analyzed using the Rotor-Gene 6000 series System Soft program version 6 (Corbett) and normalized to the expression of glyceraldehyde 3-phosphate dehydrogenase (Gapdh).

### 4.10. Statistical Analysis

The data were presented as the mean ± SEM. Statistical analyses were conducted using Statistical Analysis System for Windows version 9.4 (SAS Institute, Cary, NC, USA). The Student’s *t*-test and one-way analysis of variance were used. A *p*-value of less than 0.05 was considered statistically significant.

## 5. Conclusions

Aging is a biological process that leads to functional and structural changes in tissues and cells over time. These changes can be attributed to oxidative stress, a condition characterized by the excessive generation of chemical substances, such as free radicals, within cells. This can cause damage to the cell’s DNA, proteins, and other important biomolecules, which, in turn, can lead to inflammation. Inflammation is a major mechanism underlying many diseases associated with aging. Inflammatory responses can induce cellular apoptosis and tissue damage, especially in the context of muscles. Cellular apoptosis leads to a reduction in the number of muscle cells, which subsequently diminishes the size and function of the muscles, ultimately causing muscle atrophy and weakness. Therefore, this study focused on *Ulmus macrocarpa* Hance, a plant traditionally used to treat inflammatory conditions, among others, with recent research highlighting its excellent antioxidant and anti-inflammatory effects. This study aimed to investigate the biomarkers for antioxidant, anti-inflammatory, muscle degradation, and muscle synthesis effects of *Ulmus macrocarpa* Hance extract, renowned for its exceptional antioxidant and anti-inflammatory properties. Previous studies conducted by our research team have shown significant activity of *Ulmus* extracts against muscle loss at the in vitro stage. In this study, in vivo experiments were conducted to investigate the effects of *Ulmus macrocarpa* Hance extract (UME) on muscle-related biomarkers. During the in vivo stage, UME administration was observed to alleviate muscle loss-induced reductions in body weight and muscle mass, increase muscle fiber cross-sectional area, and enhance exercise capacity and activation of Akt, mTOR, and Foxo3α, which are important regulators of muscle protein synthesis and degradation pathways. Additionally, UME administration decreased the expression of apoptosis-related factors such as Bax and increased the expression of muscle formation-related genes like MyoD, Myogenin, and IGF-1, while decreasing the expression of muscle atrophy-related genes including Myostatin, Atrogin1, and MuRF1. Furthermore, UME administration significantly reduced the expression of inflammatory biomarkers IL-1β and IL-6, while increasing the levels of antioxidant biomarkers such as SOD, catalase, and GPx. This effectively mitigated oxidative stress induced by muscle loss. Overall, these findings suggest that UME has potential therapeutic effects against muscle loss and atrophy through multiple pathways, including promoting muscle protein synthesis, inhibiting muscle protein breakdown, reducing inflammation, and enhancing antioxidant defense mechanisms. These results provide valuable insights for the development of pharmaceuticals and health supplements targeting muscle loss and atrophy

## Figures and Tables

**Figure 1 ijms-25-06197-f001:**
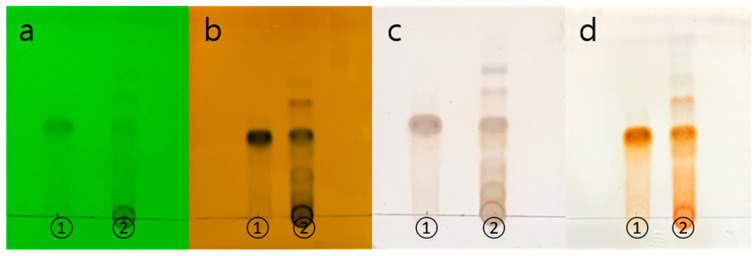
TLC chromatograms of each sample compound and references: (**a**) UV 254 nm, (**b**) FeCl_3_, (**c**) 10% H_2_SO_4_, (**d**) ρ-Anisaldehyde H_2_SO_4_. The eluent system used was chloroform/methanol/water = 70:30:4 (*v/v/v*). ① catechin 7-O-*β*-D apiofuranoside, ② *Ulmus macrocarpa* Hance extract (UME).

**Figure 2 ijms-25-06197-f002:**
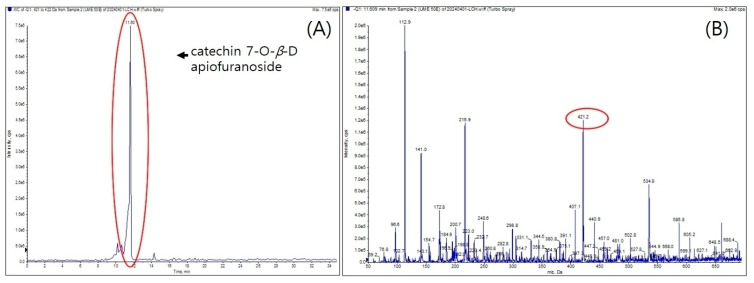
Negative mode LC-MS/MS of UME: (**A**) = extracted ion chromatogram of UME, (**B**) = total ion chromatogram.

**Figure 3 ijms-25-06197-f003:**
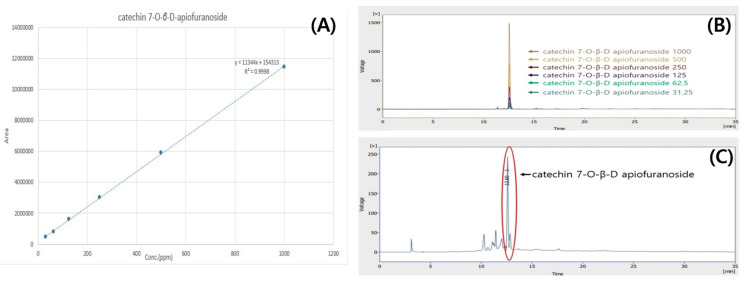
(**A**) Calibration curve and equation of 7-O-*β*-D-apiofuranoside, Y= 11,344X + 154,313 (R^2^ = 0.999); (**B**) HPLC chromatogram of 7-O*-β*-D-apiofuranoside; (**C**) HPLC chromatogram of the extract of UME.

**Figure 4 ijms-25-06197-f004:**
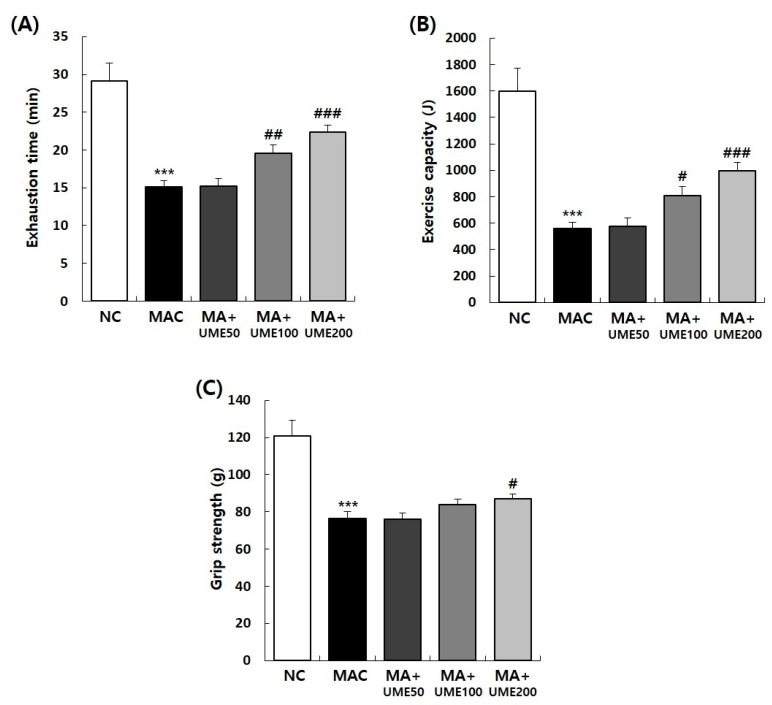
Effects of UME administration on exercise capacity and grip strength in mice with dexamethasone-induced muscle atrophy. (**A**) Endurance time to exhaustion, (**B**) exercise capacity, and (**C**) grip strength. Data are expressed as the mean ± SEM (*n* = 10). *** *p* < 0.001 indicate significant differences compared to the NC group. ^#^ *p* < 0.05, ^##^ *p* < 0.01, ^###^ *p* < 0.001 indicate significant differences compared to the MAC group.

**Figure 5 ijms-25-06197-f005:**
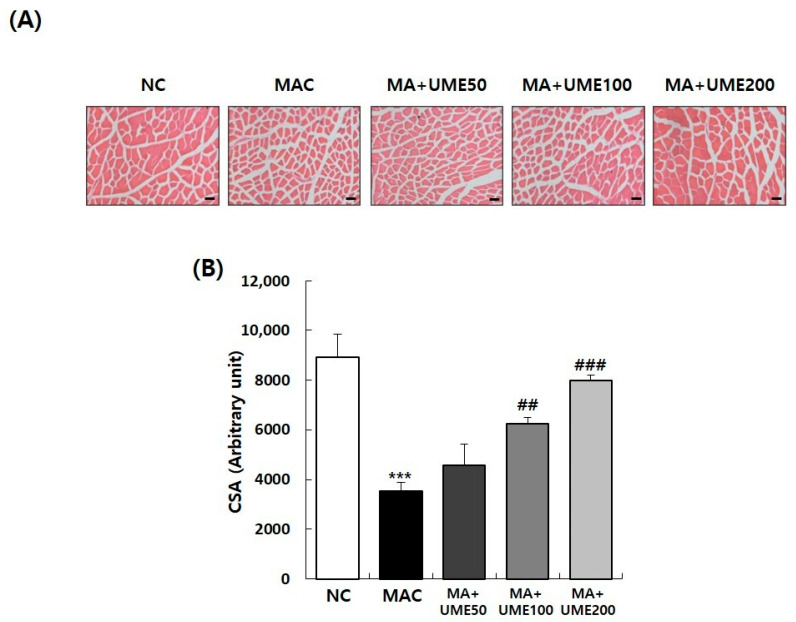
Effect of UME administration on skeletal muscle atrophy in mice with dexamethasone-induced muscle atrophy. (**A**) Representative images of tibialis anterior muscle stained with H and E. (**B**) Quantitative data on cross-sectional area of myofibers in tibialis anterior muscle. Data are expressed as the mean ± SEM (*n* = 5). The scale bar is 50 μm *** *p* < 0.001 indicate significant differences compared to the NC group. ^##^ *p* < 0.01, ^###^ *p* < 0.001 indicate significant differences compared to the MAC group.

**Figure 6 ijms-25-06197-f006:**
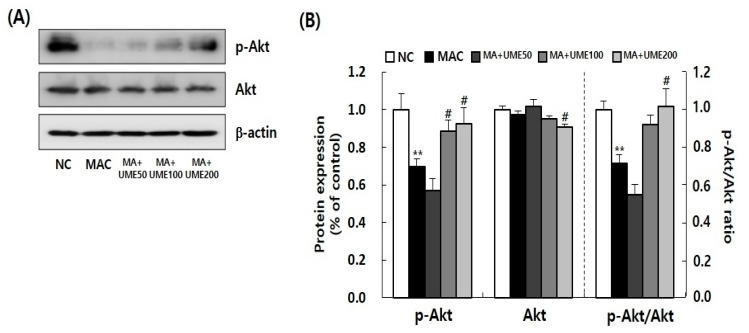
Effect of UME administration on the Akt/mTOR and FoxO3a signaling pathway in the gastrocnemius muscle of dexamethasone-induced muscle atrophic mice was examined. The protein expressions of (**A**) phospho-Akt and Akt, (**C**) phospho-mTOR and mTOR, and (**E**) phospho-FoxO3a and FoxO3a were analyzed using Western blot analysis. (**B**,**D**,**F**) Quantitative analysis of the Western blot results was conducted. Each protein expression level was normalized to that of β-actin and expressed relative to the NC group. The data are presented as the mean ± SEM (*n* = 10). ** *p* < 0.01, *** *p* < 0.001 indicate significant differences compared to the NC group. ^#^ *p* < 0.05, ^##^ *p* < 0.01, ^###^ *p* < 0.001 indicate significant differences compared to the MAC group.

**Figure 7 ijms-25-06197-f007:**
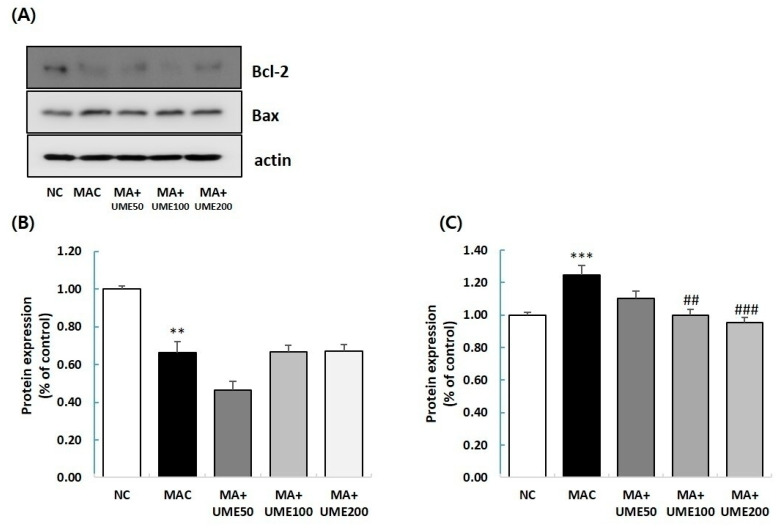
Effect on expression of apoptosis regulatory proteins. The protein expressions of (**A**) Bcl-2 and Bax, (**B**) Bcl-2, and (**C**) Bax were analyzed by Western blot analysis. Quantitative analysis of the Western blot results was performed for (**B**) Bcl-2 and (**C**) Bax. Each protein expression level was normalized to that of β-actin and expressed relative to the NC group. The data are expressed as the mean ± SEM (*n* = 10). ** *p* < 0.01, *** *p* < 0.001 indicate significant differences compared to the NC group. ^##^ *p* < 0.01, ^###^ *p* < 0.001 indicate significant differences compared to the MAC group.

**Figure 8 ijms-25-06197-f008:**
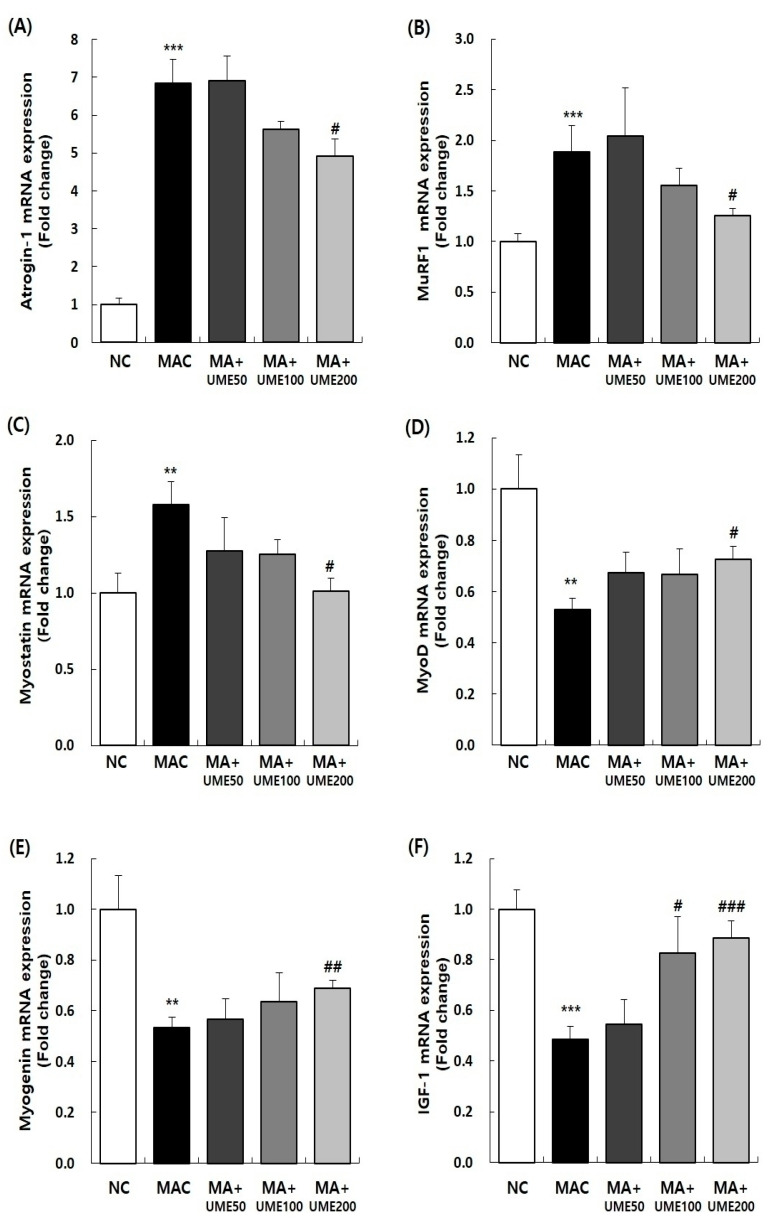
Effect of UME administration on mRNA expressions of muscle synthesis- or degradation-related genes in the soleus muscle of dexamethasone-induced muscle atrophic mice. The relative mRNA expression levels of (**A**) Atrogin-1, (**B**) Muscle Ring-Finger protein-1 (MuRF1), (**C**) Myostatin, (**D**) MyoD, (**E**) Myogenin, and (**F**) IGF-1 were analyzed by real-time PCR. The target mRNA expression was normalized to that of GAPDH. Data are expressed as the mean ± SEM (*n* = 6). ** *p* < 0.01, *** *p* < 0.001 indicate significant differences compared to the G1 group. ^#^ *p* < 0.05, ^##^ *p* < 0.01, ^###^ *p* < 0.001 indicate significant differences compared to the MAC group.

**Figure 9 ijms-25-06197-f009:**
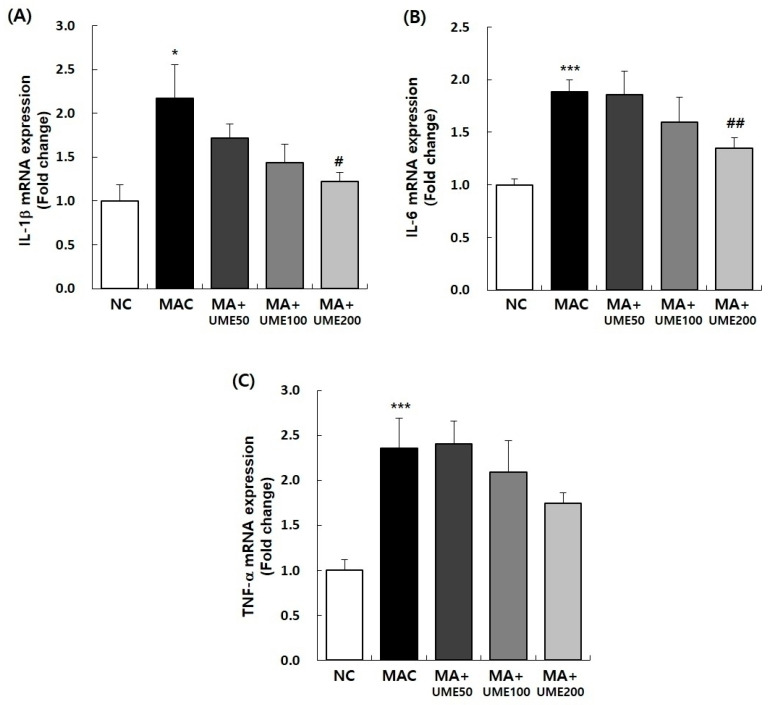
Effect of UME administration on mRNA expressions of pro-inflammatory cytokines in the soleus muscle of dexamethasone-induced muscle atrophy mice. The relative mRNA expression levels of (**A**) IL-1b, (**B**) IL-6, and (**C**) TNF-a were analyzed by real-time PCR. The target mRNA expression was normalized to that of GAPDH. Data are expressed as the mean ± SEM (*n* = 6). * *p* < 0.05, *** *p* < 0.001 indicate significant differences compared to the NC group. ^#^ *p* < 0.05, ^##^ *p* < 0.01 indicate significant differences compared to the MAC group.

**Figure 10 ijms-25-06197-f010:**
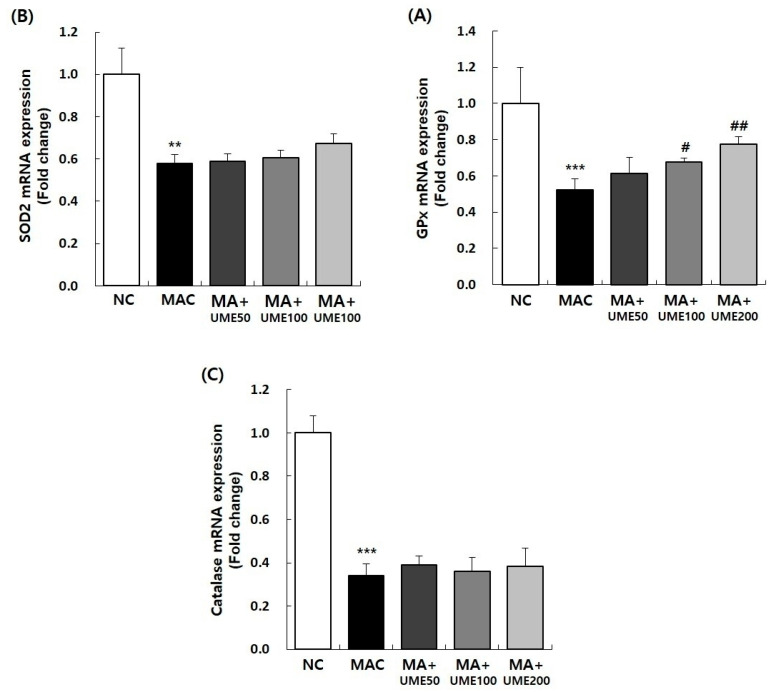
Effect of UME administration on mRNA expressions of antioxidant enzymes in the soleus muscle of dexamethasone-induced muscle atrophic mice. The relative mRNA expression levels of (**A**) GPx, (**B**) SOD2, and (**C**) catalase were analyzed by real-time PCR. The target mRNA expression was normalized to that of GAPDH. Data are expressed as the mean ± SEM (*n* = 6). ** *p* < 0.01, *** *p* < 0.001 indicate significant differences compared to the NC group. ^#^ *p* < 0.05, ^##^ *p* < 0.01 indicate significant differences compared to the MAC group.

**Figure 11 ijms-25-06197-f011:**
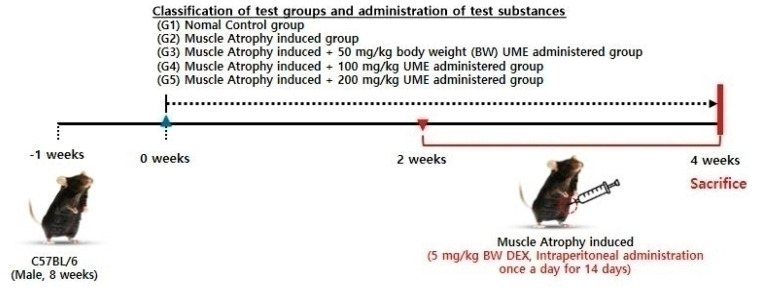
Experimental design and treatment.

**Table 1 ijms-25-06197-t001:** Body weight.

	0 Week	1 Week	2 Week	3 Week	4 Week
**G1**	23.83 ± 0.47	25.31 ± 0.52	26.13 ± 0.51	27.99 ± 0.65	28.56 ± 0.74
**G2**	23.76 ± 0.40	25.75 ± 0.42	26.35 ± 0.48	26.95 ± 0.52 *	25.56 ± 0.45 **
**G3**	23.77 ± 0.41	24.97 ± 0.34	25.77 ± 0.46	26.17 ± 0.54	24.48 ± 0.54
**G4**	23.74 ± 0.45	24.85 ± 0.60	25.50 ± 0.71	26.39 ± 0.82	25.67 ± 0.98
**G5**	23.59 ± 0.30	24.94 ± 0.48	25.84 ± 0.54	26.42 ± 0.59	24.63 ± 0.54

Values are expressed as mean ± SEM. * *p* < 0.05, ** *p* < 0.01 indicate significant differences compared to the G1 group. (G2).

**Table 2 ijms-25-06197-t002:** Body fat percentage and lean body mass percentage.

	G1	G2	G3	G4	G5
**Fat (%)**	15.29 ± 0.43	19.85 ± 0.54 *	19.74 ± 0.52	18.72 ± 0.85	18.33 ± 0.77
**Lean body (%)**	84.71 ± 0.43	80.15 ± 0.54 ***	80.26 ± 0.52	81.28 ± 0.85	81.67 ± 0.77

* *p* < 0.05, *** *p* < 0.001 indicate significant differences compared to the G1 group. (G2). Values are expressed as mean ± SEM.

**Table 3 ijms-25-06197-t003:** Muscle weight (g) and relative muscle weight (g/100 g body weight).

		G1	G2	G3	G4	G5
**Muscle weight (g)**	**QF**	0.343 ± 0.012	0.262 ± 0.008 ***	0.271 ± 0.016	0.291 ± 0.016	0.308 ± 0.013 ^##^
	**GA**	0.0.315 ± 0.011	0.249 ± 0.008 ***	0.235 ± 0.010	0.253 ± 0.012	0.255 ± 0.008
	**SOL**	0.017 ± 0.001	0.013 ± 0.001 *	0.014 ± 0.001	0.017 ± 0.001 ^##^	0.018 ± 0.000 ^###^
	**EDL**	0.025 ± 0.004	0.014 ± 0.001 *	0.018 ± 0.002	0.020 ± 0.002 ^##^	0.023 ± 0.002 ^##^
	**TA**	0.124 ± 0.005	0.089 ± 0.004 ***	0.090 ± 0.004	0.093 ± 0.005	0.096 ± 0.004
**Relative muscle w eight (g/100 g body weight)**	**QF**	1.208 ± 0.048	1.028 ± 0.038 **	1.109 ± 0.066	1.152 ± 0.080	1.251 ± 0.043 ^##^
	**GA**	1.105 ± 0.032	0.977 ± 0.037 *	0.964 ± 0.040	1.002 ± 0.069	1.035 ± 0.027
	**SOL**	0.059 ± 0.004	0.050 ± 0.004	0.058 ± 0.002	0.066 ± 0.003 ^##^	0.072 ± 0.002 ^###^
	**EDL**	0.089 ± 0.015	0.056 ± 0.002 *	0.074 ± 0.009	0.079 ± 0.008 ^#^	0.093 ± 0.009 ^###^
	**TA**	0.434 ± 0.0018	0.349 ± 0.014 **	0.370 ± 0.016	0.365 ± 0.025	0.391 ± 0.013 ^#^

Values are expressed as mean ± SEM. * *p* < 0.05, ** *p* < 0.01, *** *p* < 0.001 indicate significant differences compared to the G1 group. (G2). ^#^ *p* < 0.05, ^##^ *p* < 0.01, ^###^ *p* < 0.001 indicate significant differences compared to the G2 group. (G3, G4, and G5).

**Table 4 ijms-25-06197-t004:** Changes in mRNA expression in muscle (soleus muscle, SOL).

	G1	G2	G3	G4	G5
**MyoD**	1.00 ± 0.13	0.53 ± 0.05 **	0.67 ± 0.08	0.67 ± 0.10	0.73 ± 0.05 ^#^
**Myogenin**	1.00 ± 0.13	0.53 ± 0.04 **	0.57 ± 0.08	0.64 ± 0.11	0.69 ± 0.03 ^##^
**IGF-I**	1.00 ± 0.08	0.49 ± 0.05 ***	0.55 ± 0.10	0.83 ± 0.14 ^#^	0.89 ± 0.07 ^###^
**Myostatin**	1.00 ± 0.13	1.58 ± 0.15 **	1.28 ± 0.22	1.25 ± 0.10	1.01 ± 0.09 ^#^
**Atrogin1**	1.00 ± 0.18	6.84 ± 0.64 ***	6.90 ± 0.66	5.63 ± 0.20	4.91 ± 0.46 ^#^
**MuRF1**	1.00 ± 0.08	1.88 ± 0.26 **	2.04 ± 0.48	1.55 ± 0.17	1.25 ± 0.07 ^#^
**IL-1β**	1.00 ± 0.19	2.17 ± 0.38 *	1.72 ± 0.16	1.44 ± 0.21	1.22 ± 0.10 ^#^
**IL-6**	1.00 ± 0.06	1.88 ± 0.12 ***	1.86 ± 0.23	1.59 ± 0.24	1.35 ± 0.10 ^##^
**TNF-α**	1.00 ± 0.12	2.35 ± 0.34 ***	2.40 ± 0.26	2.09 ± 0.35	1.74 ± 0.12
**SOD2**	1.00 ± 0.12	0.58 ± 0.04 **	0.59 ± 0.04	0.61 ± 0.04	0.67 ± 0.05
**Catalase**	1.00 ± 0.08	0.34 ± 0.05 ***	0.39 ± 0.04	0.36 ± 0.06	0.38 ± 0.09
**GPx1**	1.00 ± 0.20	0.52 ± 0.06 ***	0.61 ± 0.09	0.68 ± 0.02 ^#^	0.78 ± 0.04 ^##^

Values are expressed as mean ± SEM. * *p* < 0.05, ** *p* < 0.01, *** *p* < 0.001 indicate significant differences compared to the G1 group. (G2). ^#^ *p* < 0.05, ^##^ *p* < 0.01, ^###^ *p* < 0.001 indicate significant differences compared to the G2 group. (G3, G4, and G5).

**Table 5 ijms-25-06197-t005:** Primer sequences for the real-time PCR used in this study.

mRNA	Primer Sequences	
Atrogin-1	Forward	5′-GCCCTCCACACTAGTTGACC-3′
	Reverse	5′-GACGGATTGACAGCCAGGAA-3′
Catalase	Forward	5′-GAACGAGGAGGAGAGGAAAC-3′
	Reverse	5′-TGAAATTCTTGACCGCTTTC-3′
GPx1	Forward	5′-CAGGTCGGACGTACTTGAG-3′
	Reverse	5′-CAGGTCGGACGTACTTGAG-3′
IGF-1	Forward	5′-GTGGATGCTCTTCAGTTCGTGTG-3′
	Reverse	5′-TCCAGTCTCCTCAGATCACAGC-3′
IL-1β	Forward	5′-TGGACCTTCCAGGATGAGGACA-3′
	Reverse	5′-GTTCATCTCGGAGCCTGTAGTG-3′
IL-6	Forward	5′-CCTCTGGTCTTCTGGAGTACC-3′
	Reverse	5′-ACTCCTTCTGTGACTCCAGC-3′
MuRF1	Forward	5′-GAGGGCCATTGACTTTGGGA-3′
	Reverse	5′-TTTACCCTCTGTGGTCACGC-3′
MyoD1	Forward	5′-GCACTACAGTGGCGACTCAGAT-3′
	Reverse	5′-TAGTAGGCGGTGTCGTAGCCAT-3′
Myogenin	Forward	5′-CCATCCAGTACATTGAGCGCCT-3′
	Reverse	5′-CTGTGGGAGTTGCATTCACTGG-3′
Myostatin	Forward	5′-ACTGGACCTCTCGATAGAACACTC-3′
	Reverse	5′-ACTTAGTGCTGTGTGTGTGGAGAT-3′
SOD2	Forward	5′-ATCAGGACCCATTGCAAGGA-3′
	Reverse	5′-AGGTTTCACTTCTTGCAAGCT-3′
TNF-α	Forward	5′-GGTGCCTATGTCTCAGCCTCTT-3′
	Reverse	5′-GCCATAGAACTGATGAGAGGGAG-3′
GAPDH	Forward	5′-TGGGTGTGAACCATGAGAAG-3′
	Reverse	5′-GCTAAGCAGTTGGTGGTGC-3′

## Data Availability

The original contributions presented in the study are included in the article, further inquiries can be directed to the corresponding author.

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
