# Peer review of "The Impact of Ulmus macrocarpa Extracts on a Model of Sarcopenia-Induced C57BL/6 Mice"

_ijms, 2024, doi:10.3390/ijms25116197_

Round 1

Reviewer 1 Report

Comments and Suggestions for Authors

In this manuscript, the author investigated the anti-inflammatory and antioxidant properties of U. macrocarpa in muscle atrophy mice models. The manuscript is at an elementary level, though addresses research questions. I consider the topic original or relevant in the field and address a specific gap. I have some comments for the author.

  1. The author should follow the plant nomenclature style Ulmus macrocarpa 
  2. Reference is missing at line 121.
  3. Did the author quantify the amount of catechin 7-O-β-D-apiofuranoside U. macrocarpa extract?
  4. Did the author perform any toxicity study with this extract?
  5. The discussion needs to be on catechin 7-O-β-D-apiofuranoside and its biological properties according to this study.
Comments on the Quality of English Language

Extensive revision of the English language is necessary.

Reviewer 2 Report

Comments and Suggestions for Authors

Authors have described the 1. myogenic, 2. anti-oxidative stress, 3. anti-apoptotic and 4. 'inhibition of muscle protein breakdown' properties of Ulmus Marcrocarpa Hance extracts (UME) in an in vivo mouse model of atrophy.  

They have conducted a robust assessment of the signaling mechanisms that support each of the four categories mentioned above.  

Minor comments

1. A schematic to depict the signaling mechanisms they have investigated for each aspect of sarcopenia pathogenesis will be beneficial to the readers.  

2. Authors have not provided a rationale or background for investigating IGF-1 in the introduction. Please provide references to support IGF-1 and its myogenic potential in the introduction.

3. In Table 3, is the amount of exercise the same as 'exercise capacity' described in Methods? If yes, authors should maintain consistent terminology for describing 'amount of exercise' through the manuscript.

4. Figure 6B should read CSA and not CAS on the Y-axis for cross-sectional area.

5. Line 436 of the manuscript can be rephrased to something along the lines of 'the decrease was mitigated...'. Currently, the phrasing seems confusing to read.  

6. Figure 7E. Authors can include a better representative western blot image of FOXO3a protein differences in MAC versus treatment groups.  

7. Figure 8B and 8C require axes. This figure seems incomplete. Are the significance asterices missing in this plot? Also, better representative western blot images for Bax can be added to show decrease in expression with treatment

Reviewer 3 Report

Comments and Suggestions for Authors

In the article entitled "The Impact of Ulmus Macrocarpa Extracts on a Model of Sarcopenia-induced C57BL/6 Mice" authors focused on the anti-inflammatory, antioxidant and antiatrophic properties of Korean Ulmus macrocarpa Hance in GC-induced muscle atrophy model.

The authors demonstrate that Ulmus extract administration promoted genes related to muscle formation while reducing those associated with muscle atrophy. In addition the authors convice that Ulmus extract mitigate inflammation and boost muscle antioxidants.
In general, the experiment has been well planned, according to the universally accepted principles in this matter, and neither the selection and size of the groups nor most of the measurement methods used are questionable, and deserve to be commended.
Nevertheless, the article suffers from a fair number of more or less major linguistic and factual errors and needs to be improved on these points.
Major concerns:

1) In the model used in the experiment, the authors administered a relatively high dose of dexamethasone. Please explain and possibly include in the manuscript why the choice of just 5mg/kg BW instead of the commonly used 2mg/kg BW. Dexamethasone in long-term administration is extremely devastating, including the gastrointestinal tract.

This raises a question and a doubt: why was there no monitoring of the amount of feed consumed by the animals?

Perhaps the weight of the body (and, by extension, the weight of individual muscles) is the resultant of the reduction in consumption by the animals, rather than the atrophy induced by DEX per se? Specifically, weight measurements were taken extremely infrequently, at a 1-week interval. For the future, I would suggest weight monitoring with at most a 2-day interval.

2) I'm not a big proponent of determining oxidative stress parameters in the blood (such as glutathione levels or antioxidant enzyme activity) if the target tissue being tested is skeletal muscle, for example. Measurements of these markers in the blood correlate relatively poorly with what may be happening at the level of skeletal muscle, hence the formulation used by the authors , cited here: “Ulmus extract (...) boosted muscle antioxidants” is too far-fetched speculation not supported by the results that were obtained (we only have increased expression for GPx).

Minor concerns:

Line 73: I don't think enzyme names need to be spelled with a capital letter

Line 188-206: please include information on the age and initial weight of the animals (admittedly, the age is given in the figure 1, but the text should also include such information)

In addition, please explain why it was decided to precede the induction of atrophy with 2 weeks of initial supplementation of Ulmus extract.
Line 190: rats?

Line 194: what volume of solution was administered

Line 198: production of which company dex was administered

Line 242-250: Please list the concentrations of primary and secondary antibodies used, along with the catalog numbers of the antibodies. This is an obligation.

Line 242: How many percent homogenate was used for the determination. How was the homogenate prepared?

Line 307/308/358/398/460/484/525 and many others: please use the description of statistical significance only as used in a given table or figure. Otherwise it leads the reader into unnecessary information chaos.

Table 3 (and some others figures and tables): I don't quite understand comparing only G2 vs. G1, without analyzing the significance of the differences between the different groups, e.g. G1 vs. G3 (as in the aforementioned Table 3).

From my perspective, the analysis of the efficacy of the substance used should not only show superiority over the placebo-induced atrophy group, but allow us to assess the strength of the effect with respect to full homeostatic conditions (in this case the G1 group).

Figure 5/Table 3: The authors should decide whether they present the results in figure or table form. I see no need to duplicate the content.

Line 345: It is not entirely clear to me how the body composition components were measured. In the methods section, I see no mention of the apparatus used (?).

Line 359: I think the word Muscle should be written with a capital letter

Round 2

Reviewer 1 Report

Comments and Suggestions for Authors

The authors have satisfactorily responded to all comments and made the necessary changes to the manuscript.